# Neurogenic Lower Urinary Tract Dysfunction in Spinal Dysraphism: Morphological and Molecular Evidence in Children

**DOI:** 10.3390/ijms24043692

**Published:** 2023-02-12

**Authors:** Dafni Planta, Tim Gerwinn, Souzan Salemi, Maya Horst

**Affiliations:** 1Division of Pediatric Urology, University Children’s Hospital Zurich, 8032 Zurich, Switzerland; 2Children’s Research Center, University Children’s Hospital Zurich, 8032 Zurich, Switzerland; 3Laboratory for Urologic Oncology and Stem Cell Therapy, Department of Urology, University Hospital Zurich, 8091 Zurich, Switzerland

**Keywords:** neurogenic lower urinary tract dysfunction, spinal dysraphism, myelomeningocele, detrusor–sphincter dyssynergia

## Abstract

Spinal dysraphism, most commonly myelomeningocele, is the typical cause of a neurogenic lower urinary tract dysfunction (NLUTD) in childhood. The structural changes in the bladder wall in spinal dysraphism already occur in the fetal period and affect all bladder wall compartments. The progressive decrease in smooth muscle and the gradual increase in fibrosis in the detrusor, the impairment of the barrier function of the urothelium, and the global decrease in nerve density, lead to severe functional impairment characterized by reduced compliance and increased elastic modulus. Children present a particular challenge, as their diseases and capabilities evolve with age. An increased understanding of the signaling pathways involved in lower urinary tract development and function could also fill an important knowledge gap at the interface between basic science and clinical implications, leading to new opportunities for prenatal screening, diagnosis, and therapy. In this review, we aim to summarize the evidence on structural, functional, and molecular changes in the NLUTD bladder in children with spinal dysraphism and discuss possible strategies for improved management and for the development of new therapeutic approaches for affected children.

## 1. Introduction

Spinal dysraphism refers to developmental anomalies caused by a lack of closure of the fetal spinal cord, which leads to anatomical abnormalities and neurological impairment of varying degrees. In children, spinal dysraphism is the most common cause of neurogenic lower urinary tract dysfunction (NLUTD), and its most common manifestation is myelomeningocele (MMC). Other forms of spinal dysraphism include closed dysraphism with tethered spinal cord and/or sacral agenesis [1].

In NLUTD, the bladder loses the ability to store and fully empty urine at low pressure either partially or completely. Although there are similarities between NLUTD in adults and children, the disease presents a unique challenge in children due to their developing and growing organism [2,3]. Spinal dysraphism occurs in the fetal period, typically between third and fourth week of gestation [3]. Therefore, bladder innervation is already impaired during fetal development, and it has been shown that structural changes in the bladder wall occur as early as the 16th to 21st week of gestation in human fetuses with MMC [4]. The bladder wall structure and function are clearly impaired in children with MMC [4,5,6]. Among the factors contributing to bladder wall impairment are progressive deterioration of nerves during pregnancy, developmental malformation of the bladder, and pathophysiological changes due to neurogenic detrusor overactivity (NDO), detrusor sphincter dyssynergia (DSD), or inflammation [7,8]. However, the exact pathogenesis of NLUTD remains unclear.

Impairment of the central (CNS), peripheral (PNS), and autonomic (ANS) nervous system can result in NLUTD. Unlike in adults with spinal cord injury (SCI), the extent of neurological lesions correlates poorly with the severity of NLUTD in congenital spinal dysraphism [1]. Furthermore, NLUTD can be functionally assigned to one of four types, based on the interaction of the detrusor and sphincter muscles, where both the detrusor and the sphincter can be either overactive or underactive [9]. In particular, an overactive and permanently contracted sphincter, as well as an uncoordinated interaction between the sphincter and detrusor, the so-called detrusor–sphincter dyssynergia, cause a functional bladder outlet obstruction (BOO). The consequences of BOO in adults are detrusor hypertrophy to overcome the functional subvesical obstruction. If untreated, the continuous remodeling of the bladder wall leads to fibrosis and eventually decompensation with chronic overdistension, diminished voiding function, and increased post-void volume [10].

Progressive changes lead to severe functional impairment with a loss of bladder wall elasticity and high intravesical pressure [6]. Most children with MMC are born with a normal upper urinary tract, but many develop a hostile bladder in early childhood with the consequences of hydronephrosis, vesicoureteral reflux, recurrent urinary tract infections, and chronic renal failure, despite early proactive therapy [2,11]. 

Our knowledge about the structural and molecular changes in NLUTD originates from animal studies and from tissue analysis of humans with NLUTD (mainly from adults after SCI), and of bladders affected by BOO (mainly from adults with benign prostatic hyperplasia). However, little is known about the effects of impaired bladder innervation in the fetus and during childhood. Analyses of fetal human bladders from fetuses with MMC are scarce and date from the end of the last century [4,12,13]. More recent animal studies investigating induced spinal dysraphism revealed similar structural bladder changes as described above, including neural, muscular, and stromal dysplasia in the bladder at an early gestational age [14,15,16]. 

Understanding the pathophysiology of congenital NLUTD is a prerequisite to improving management and to developing new treatment strategies for affected children. Therefore, the aim of the present study is to review the available evidence on structural, functional, and molecular changes in NLUTD of children with congenital spinal dysraphism (Figure 1).

## 2. Structural and Morphological Properties

Among the different factors potentially contributing to the changes in the bladder wall of children with MMC are the impaired innervation itself as well as the developmental and functional changes resulting thereof [7,8,14]. Significant structural changes in the bladders of people with MMC have been described in previous studies [4,5,6]. Most children with MMC are born with a normal upper urinary tract, but with the progressing NLUTD, the majority of these children develop bladder wall thickening with trabeculation and increased collagen deposition in the extracellular matrix (ECM) in early childhood. Bladder wall fibrosis leads to stiffening of the bladder wall and, in the long-term, to low compliance, which increases intravesical pressure and exacerbates renal injury [2,11]. This chapter will discuss the microscopic and macroscopic structural properties of NLUTD caused by spinal dysraphism. 

### 2.1. Detrusor

The urinary bladder is a hollow organ consisting of the detrusor and a luminal mucosal membrane, which is composed of the lamina propria and the urothelium. The detrusor consists of 60–70% smooth muscle cells (SMCs) embedded in the ECM and represents the main component of both the reservoir and voiding function [17]. The spindle-shaped SMCs are highly elastic cells, which stretch during filling and contract during voiding [17]. The detrusor consists of three layers; the inner and outer layers consist of longitudinally arranged muscle fibers, while muscle fibers in the middle layer are arranged in a circular pattern [18,19]. The ECM predominantly consists of collagen types I and III, which are produced by fibroblasts in a ratio of 3:1 [20]. With their tensile and compliance properties, collagens have an important mechanical function and enable filling at low pressure [20]. Further components of the ECM are elastin, providing passive recoil strength to the bladder wall during voiding, and fibronectin, guiding cell attachment and organizing the ECM. Furthermore, proteoglycans as a base material help the bladder wall to withstand high compressive forces [19]. The composition of the ECM is dynamic, and its constant interaction with the SMCs strongly influences their proliferation, differentiation, and homeostasis [21].

Studies in human fetuses showed the presence of SMCs in the bladder wall as early as gestational weeks 7–11, with an increase in thickness of the muscle layer throughout gestation [22,23]. Functional studies in the bladder of fetal sheep and calves showed that the detrusor is fully developed by the end of the second trimester, but functional innervation increases steadily towards the end of pregnancy [24,25]. In children with MMC, structural changes in the bladder can be observed already in utero. In a histological analysis of tissues derived from human fetuses, Shapiro et al. showed lower detrusor smooth muscle differentiation and a significant increase in intra- and interfascicular ECM compared to controls at 20 weeks of gestation [12]. Liu et al. demonstrated that the proliferation of SMCs was inhibited in fetal MMC mice compared to controls [7]. This indicates that in congenital NLUTD, a developmental abnormality of the bladder wall already exists in utero and contributes to the postnatal course of bladder dysfunction [7]. 

The hostile NLUTD bladder in children with MMC is typically thickened and trabeculated, has a small capacity, and has low compliance [1,4]. Compared to children and adults with normal bladder function, the smooth muscle content of the NLUTD bladder tissue is significantly reduced, with scarce smooth muscle bundles and decreased bundle sizes, while the amount of connective tissue is significantly increased [4,6]. The smooth muscle loss is compensated by a buildup of type III collagen, resulting in a greater type III/type I collagen ratio [6,26,27,28,29]. Collagen type I offers high tensile strength and rigidity, while collagen type III increases tissue flexibility and distension [30]. It is suggested that the collagen type I/type III ratio within collagen fibrils determines the passive biomechanical tissue properties [27]. Moreover, it was reported, that collagen synthesis of fibroblasts is altered by mechanical stress [31], thus, possibly leading to bladder wall fibrosis and ECM buildup in NLUTD due to functional BOO.

The early NLUTD shows dyscoordination of the detrusor and the external urethral sphincter, incomplete voiding, and increased intravesical pressure [2,8,11]. Sillen et al. clinically investigated the development of bladder dysfunction in infants with MMC by analyzing their urodynamic patterns [11]. Bladder outlet obstruction was found at the one-month assessment in 32%, and 73% of these showed NDO. These numbers slightly increased during the first year of life [11]. In a recent study by Alatas et al., DSD was present in 92% and NDO in 59% of neonates with MMC before surgical closure [32]. Even in older individuals with MMC, a prevalence of 56% of DSD was shown by Yu and Kuo [33]. It must be assumed that DSD plays a crucial role in the changes in the bladder wall in NLUTD already in early childhood, analogous to the BOO of other etiologies. The remodeling process in adults with BOO caused by benign prostatic obstruction was described in three phases by Fusco et al. [10]. Initially, the increased intravesical pressure leads to SMC hypertrophy and proliferation as a compensatory mechanism to overcome the increased bladder outlet pressure. This is enhanced by focal pressure-induced hypoxia, which stimulates blood flow in the bladder wall [10,34]. The bladder may remain in a compensated state for a prolonged period of time until a persistent obstruction leads to decompensation, resulting in the loss of smooth muscle and neuronal cells, as well as in an increased and changed ECM [10]. Bladder wall fibrosis caused by mechanical stress is characterized by abnormal deposition of connective tissue through fibroblastic and cytokine-mediated inflammatory response, and it causes the loss of normal detrusor contractility and compliance of the bladder [35]. 

### 2.2. Urothelium and Lamina Propria

The urothelium is the innermost layer of the bladder, facing the bladder lumen and constituting a barrier to blood–urine permeability [36]. It consists of multiple layers of basal, intermediate, and umbrella cells. The umbrella cells in the superficial layer produce uroplakins that form a urine-proof and shielding plaque [37,38]. The urothelium has specialized sensory and signaling properties and plays an important role in afferent signaling. The urothelial cells possess pain- and mechanoreceptors and they release both signaling molecules and growth factors, which affect sensory nerve functions [20,39,40]. The underlying lamina propria is a submucosal connective tissue layer with blood vessels and nerves supplying the urothelium. It consists of different cell types, including afferent and efferent nerve endings, which are embedded in a network of elastin, type I and III collagen [20]. The lamina propria is also thought to have an important role in signal transduction between the detrusor muscle, the urothelium, and the central nervous system. Furthermore, it seems to be responsible for growth factor production, affecting both adjacent compartments, the overlying urothelium, and the underlying SMCs [20]. The cells in the lamina propria that may be responsible for the interaction between smooth muscle and neuronal cells are the interstitial cells of Cajal (ICC) [41,42,43]. The ICCs, located just below the urothelium, are in close contact with mucosal nerves and myofibroblasts [41,42]. Alterations in the number and distribution of ICCs have been described in various bladder diseases, however without their exact function being known [44,45]. It has been suggested that ICCs play a role in overactive bladder issues [41]. In neurogenic detrusor overactivity, a shift towards a fibroblast phenotype has been observed [42]. Furthermore, increased gap junction formation in the suburothelium was demonstrated, and it is hypothesized that this change could play a significant role in the sensory and motor pathway of the detrusor abnormality, as the mucosa may take over the neuronal role [46]. Fetal rats with induced MMC were found to have a significantly reduced number of ICCs, and it was hypothesized, that the observed bladder dysfunction may be related to the reduced density of ICCs in the developing bladder [47]. The only study performed on children with MMC and NLUTD could not confirm changes in ICC density compared to controls [48]. However, there are distinct concerns with regard to the results presented, due to the composition of the control group. Ten out of sixteen children in the control group had a combined heart anomaly and eight out of sixteen had life limiting organ dysfunction leading to their demise within the first 72 h after birth [48]. Furthermore, no information on the gestational age of the control group is given, which could lead to a very relevant bias of the results (i.e., no statistical differences). According to Shapiro et al., the urothelium in human fetal MMC bladder tissue is highly differentiated at 20 weeks gestational age [12]. To our knowledge, the only study investigating the bladder mucosa in individuals with MMC found chronic inflammation of the urothelium and the lamina propria with lymphocyte and plasma cell infiltration, disruption of the superficial epithelium with loss of uroplakin expression, an altered urothelial proliferation, squamous cell metaplasia, and, rarely, mucosal metaplasia [49]. Possible causes for these changes, such as impaired bladder innervation, increased bladder pressure, and repeated injury to the mucosa following catheterization or bacteriuria, were discussed. The changes are most likely resulting from continued trauma of the urothelium, but the exact pathogenesis remains unclear [49]. In contrast, Wu et al. studied samples from adults with NLUTD and found chronic bladder inflammation, increased apoptosis, and decreased barrier function, independent of urinary tract infections [50] (see Section 4.2). These results rather suggest that bladders in people with SCI primarily have a regenerative dysfunction, which in turn promotes bacterial infections [50].

### 2.3. Innervation

The neuronal control of the lower urinary tract and micturition is controlled through a complex network of autonomic and somatic nerves. Detrusor and smooth muscle of the internal urethral sphincter are innervated via sympathetic and parasympathetic fibers, while the striated muscle of the external urethral sphincter is supplied by the somatic pudendal nerve. Numerous central pathways that transmit information between the brain and the spinal cord contribute to the regulation of the lower urinary tract. This complex topic is presented comprehensively elsewhere [51,52].

Two types of afferent nerves in the bladder wall mediate the sensation of bladder filling; the myelinated Aδ-fibres, which are located in the detrusor layer of the bladder, are activated by low-intensity stimuli and transmit normal filling sensations, while the unmyelinated C-fibers, which are located in all bladder wall compartments, are activated by high-intensity stimuli, such as a maximum stretch of the bladder, or in pathological situations when the functional or anatomical capacity of the bladder is reduced [40]. In the fetus, urine is drained from the bladder by non-neural mechanisms. During CNS maturation in the fetal period, the micturition is initiated by primitive pathways on the spinal level. After birth, the CNS undergoes continuous development and ultimately organizes urination through the modeling influence of the higher brain regions [52]. Reflex voiding in infants is accompanied by synchronized but incomplete sphincter relaxation [52]. 

The SCI of the upper motor neuron type initially leads to a spinal shock, leaving the detrusor areflexic before a primitive spinal reflex micturition pathway reemerge and the detrusor becomes overactive. Opposed to reflex voiding in infants, the activity of the external urethral sphincter persists after suprasacral injuries, and the events lead to a simultaneous tonic activity of both urethral sphincters and the detrusor [52,53]. The structural anomalies observed in neurogenic detrusor overactivity after traumatic or degenerative SCI are a loss of global innervation with a simultaneous increase in the nerve endings in the lamina propria, a global presence of inflammation peaking in the lamina propria, fibrosis, and SMC dysfunction [54]. Clinical and experimental studies suggest that the functional properties of the bladder wall C-fibers are altered following SCI and might be partly responsible for inducing detrusor overactivity [53,55,56,57]. As urothelial cells are involved in afferent signaling, it has been hypothesized that the non-neuronal release of neurotransmitters from the urothelium plays a role in C-fiber activation in overactive bladders in children [40].

Shapiro et al. described a decreased density of neurons within the detrusor in human fetuses with MMC [12]. Haferkamp et al. analyzed the morphological detrusor innervation changes in adolescents with MMC and found axonal degeneration and depletion of terminal organelles as well as increased width of the neuromuscular junction, restricted axonal regeneration and activated Schwann cells [26,58]. Animal studies investigating the changes occurring in bladder innervation in spinal dysraphism during development are in line with these results, showing a decreased nerve density, inhibition of proliferation, and promotion of apoptosis of neuronal cells within the detrusor muscle [7,14]. According to our knowledge, no studies investigating the nerve endings within the lamina propria of MMC bladders exist so far. 

In summary, the structural changes in the bladder wall in spinal dysraphism already occur in the fetal period and affect all bladder wall compartments. The progressive decrease in smooth muscle and the gradual increase in fibrosis in the detrusor, the impairment of the barrier function of the urothelium, and the global decrease in nerve density lead to severe functional impairment. The bladder consequently loses the ability to store and release urine in a regulated manner. These findings suggest that NLUTD in spinal dysraphism is not a result of simple neurodegeneration but of a complex and chronically progressive neurogenic developmental disorder [7].

## 3. Functional Properties

The urinary bladder is a complex organ, whose two main functions are the storing of urine at low intravesical pressure and the voiding without residual urine volume. The biomechanical properties defining a healthy bladder wall are, therefore, a high elasticity required for filling, which is defined by the elastic modulus and compliance, and a strong detrusor contraction required for emptying. The bladder function is, thus, determined by the passive biomechanical properties of the bladder wall and by the active contractility of the smooth musculature of the detrusor. Bladder function is controlled by intricate circuits involving multiple neural centers at the spinal and supraspinal levels (see Section 2.3). 

Compliance and elastic properties are both largely determined by the composition of the ECM (see Section 2.1). At the initiation of smooth muscle contraction, an intracellular increase in Ca^2+^ concentration and the formation of a complex of Ca^2+^ and calmodulin lead to the ATP-dependent phosphorylation of myosin and a contraction via the sliding of actin past myosin filaments [18]. Contraction can be triggered by the depolarization of the cell membrane and by the opening of voltage-sensitive Ca^2+^ channels. Alternatively, a voltage-independent pathway works through neurotransmitters interacting with ligand-gated Ca^2+^ channels in the cell membrane. Hormones, such as estrogen, interact with specific cell membrane receptors and increase the intracellular second messenger inositol trisphosphate (IP3), inducing Ca^2+^ release from the sarcoplasmic reticulum [18].

It is important to note that the healthy detrusor is composed differently in children than in adults, resulting in different functional properties. In children, the proportion of smooth muscle in the detrusor is smaller, while the proportion of connective tissue is comparatively bigger, resulting in a greater mechanical stiffness of the bladder wall [59]. Another important difference is the lower innervation (nerve density), as evidenced by a lower contraction in response to electric field stimulation compared to adults [59]. However, both Ca^2+^ influx and maximum contractile force exerted by drug-induced contraction are identical in the detrusor tissue of children and adults [59].

### 3.1. Biomechanical Properties

The NLUTD bladder partially or completely loses its ability to store urine at low pressure and to void urine without residual volume. As described above hostile NLUTD due to detrusor overactivity and/or BOO can lead to fibrosis of the bladder wall. The biomechanical consequences are low compliance and an increased elastic modulus of the bladder wall [19]. Clinically, this results in reduced bladder capacity, high intravesical storage pressure even when the bladder is barely filled, and incomplete voiding [19]. The functional sequelae of detrusor fibrosis are similar regardless of the underlying disease, which can be SCI [60], BOO [61], or spinal dysraphism [6] (see Section 2.1). Johal et al. were able to show significantly increased detrusor tissue stiffness, with a significantly higher elastic modulus in hostile bladders of both children with MMC and children with congenital BOO caused by posterior urethral valves compared to pediatric controls [6,61]. 

### 3.2. Detrusor Contractility

Bladder wall contractility can be assessed in vivo by urodynamic measurements or ex vivo with an organ bath. An organ bath is a chamber in which drugs can be administered to isolated tissue under physiological conditions, or electrical stimuli can be applied in order to measure functions. Corresponding experimental set-ups and methods are presented in detail elsewhere [6,59,61,62,63,64].

Urinary bladder tissue of adults with normal bladder function shows constant detrusor contraction in response to direct agonists in vitro, and constant nerve-mediated excitability independent of age or gender [60]. To our knowledge, the corresponding in vitro experiments on detrusor contractility in various pediatric developmental age groups have not been performed yet.

The NLUTD in adults resulting from SCI is characterized by a decline in nerve-mediated contractions and a decreased contractile agonist-induced contractility with increasing age compared to controls [60]. Analysis of detrusor tissue samples from 18 adults with NLUTD following SCI revealed a longer time to peak contraction as well as significantly lower maximum contractile force after electrical field stimulation as a sign of functional denervation. This was in line with a decreased contractile strength in response to direct contractile stimulation with the agonist carbachol [65]. In contrast, detrusor tissue samples from adults (mean age 33 years) with neurogenic detrusor overactivity caused by SCI or multiple sclerosis exhibited similar maximum tension values in response to electrical field stimulation as a cohort of significantly older individuals with stable bladders (mean age 56 years). However, heterogeneous age, different pathologies, and notable tension value differences could be confounding variables [63]. Bladder tissues from adults affected by BOO are characterized by age-dependent functional denervation with preserved detrusor contractility after stimulation by direct agonists [60]. The analysis of 40 adults with BOO showed age-dependent functional denervation, causing a decline in nerve-mediated detrusor contractions. However, this decrease could not be attributed to impaired contractility of the detrusor muscle alone, as the contractile response to both the direct electrical field and carbachol stimulation was similar to control bladder tissue [60]. It is unknown whether the age or the persistence of the pathology in non-congenital NLUTD or BOO is the significant aspect [60]. 

Similar to NLUTD in adults, in vitro assays with tissues of children with congenital spinal dysraphism/MMC showed a reduced contractile response to both electrical field nerve-mediated stimulation and direct contractile agonist stimulation. Based on the currently available literature, no conclusions can be drawn about the relative severity of biomechanical bladder dysfunction in these children in comparison to adult tissue. Johal et al. compared the biomechanical properties of detrusor tissue samples derived from children with NLUTD to detrusor tissue biopsies from pediatric controls [6]. They analyzed nerve-mediated contraction after electrical field stimulation and contractions induced by the agonist carbachol or α, β methylene ATP. The contraction induced by the electrical field stimulation was significantly lower in NLUTD, again indicating functional denervation [6]. As shown in previous studies with adult NLUTD tissue, the contractile response to the agonist carbachol and α, β methylene ATP was also significantly lower in neurogenic detrusor compared to healthy controls [6]. A sub-analysis showed significantly reduced contraction induced by electrical field stimulation relatively compared to the agonist carbachol-induced contraction. This strongly indicates functional denervation in neurogenic samples [6]. Electrical field and agonist-induced contractions could be elicited in all healthy samples. Interestingly, four NLUTD samples showed no contractile response to electrical field stimulation, while these samples still responded to the contractile agonists, similarly to all other samples. This implies complete denervation of these four specific NLUTD detrusor samples [6]. Likewise, Gup et al. found that the maximum contractile force in response to KCL and carbachol was lower in children with MMC-induced NLUTD tissue samples than in control bladders [13].

### 3.3. Smooth Muscle Cell Contractility 

The contraction of the bladder wall is determined by the contractile capacity of the single SMCs and, eventually, by their dense and interconnected network. The SMC contraction can be measured in vitro by micropillars (for the evaluation of single smooth muscle cell contractions) [66], by collagen gel contraction assays (for the evaluation of SMC networks) [67], and in theory with an organ bath (similarly to native detrusor tissue for in vivo-like tissue-engineered bladder wall substitutes). However, Kropp et al. were able to show that isolated and in vitro cultured bladder SMCs derived from children with normal bladder function lost their contractile response to the potent contractile agonist carbachol and KCL [68]. Thus, it seems difficult to study the contractility of SMCs in vitro at the cellular level. This may be because, in a non-physiological environment, the isolated cells change their response to contractile stimuli as well as their contractile properties. However, Lin et al. were able to demonstrate a decreased contractile response in cultured NLUTD SMCs using a collagen contraction assay [69]. This shows that the cultured SMCs originating from children with MMC maintained their impaired functional characteristics in vitro [69], implying that the altered intracellular contractile machinery is present in NLUTD SMCs. Furthermore, these results are consistent with in vitro tissue analysis (see Section 3.2) and in vivo assessment by urodynamic studies, where NLUTD bladders generated a lower contractile force than controls [11]. 

In summary, the biomechanical properties of NLUTD in children with spinal dysraphism are characterized by reduced compliance and increased elastic modulus. The contractility of the bladder wall is markedly reduced. This is in line with observations made in adults with NLUTD. To our knowledge, detrusor contractility during the fetal period and throughout childhood has not yet been studied.

## 4. Molecular Changes 

Bladder function is regulated by the synergistic action between the detrusor, neurons, the mucosal layer, and the connective tissue of the bladder wall. Structural changes in the NLUTD bladder wall of people with MMC comprise inflammatory infiltration, fibrosis [26,27,70], and altered collagen distribution with a greater type III/type I collagen ratio [27] (see Section 2.1). Investigations into the molecular mechanisms underlying bladder-related diseases in children with spinal dysraphism are very challenging; the limited availability of human tissue samples, especially from children, is also due to ethical issues, and the insufficient number of samples does not allow for a thorough statistical evaluation.

### 4.1. Smooth Muscle Cells

Bladder SMCs can be efficiently isolated by explant [71] or enzymatic digest techniques [72] from human and rodent bladders. Mature SMCs isolated from healthy sources have displayed limited proliferation capacity in vitro and a loss of the contractile phenotype during in vitro expansion [68,73]. However, clear differences have been shown between SMC cultures isolated from healthy and NLUTD bladders, with the latter retaining disease-specific features and functional differences even in vitro [69]. To identify genes responsible for the abnormal phenotypes, gene expression levels of cultured SMCs from normal and NLUTD bladders were compared using cDNA gene arrays [74]. The genes and corresponding signaling pathways identified include fibroblast growth factor, PTEN signaling, and integrin signaling.

To create a genetic signature of NLUTD SMCs, Hipp et al. compared microarray analyses of SMCs derived from people with MMC to SMCs derived from able-bodied people [5]. They confirmed that MMC bladders show an excessive ECM deposition, exhibit improper contraction, and are developmentally immature compared to healthy SMCs [5]. They also identified several over- and under-expressed genes in MMC bladder SMCs [5]. Among the over-expressed genes involved in development were the mesenchyme homeobox 2, bone morphogenic protein 6, fibroblast growth factor 2, cartilage oligomeric matrix protein, and collagen (type 1A1, type 5A2, and type 11A). Downregulated genes were related to muscle contraction, cell–cell adhesion pathways, and transmission of nerve impulses. It has been demonstrated that SMCs can change from a normal contractile to a so-called synthetic or proliferative phenotype [28]. This process is characterized by changes in the expression of contractile proteins, such as smooth muscle α-actin (SMA), myosin heavy chains SM-1 and SM-2, calponin, caldesmon, SM-22a, smoothelin and myosin heavy chain 11 (MYH11) [28]. The main lineage-specific markers for the contractile SMC phenotype are α-SMA (as an early marker), calponin (as an intermediate marker), and smoothelin and MYH11 (as late markers). α-SMA is a common marker, while calponin, smoothelin, and MYH11 are highly specific to contractile and functional SMCs [28]. α-SMA and MYH11 are both involved in the sliding filament mechanism to generate contractile force in smooth muscle cells, with MYH11 being the most mature and distinct marker for contractile functional SMCs [28]. Calponin is a calcium-binding protein and modulator of myosin ATP activity [28]. It also regulates actin/myosin interaction and is phosphorylated in intact smooth muscle in response to contractile stimuli [28,75]. Smoothelin is a possible regulator of the contractile apparatus and a component of the cytoskeleton that colocalizes with actin stress fibers [28]. Smoothelin appears to reflect the contraction potential of smooth muscle. In overactive human bladders, upregulation of smoothelin has been demonstrated [76]. The importance of contractile gene and protein expression in the context of morphological and functional alterations in NLUTD in children was also confirmed by our group [77]. Structural changes, the loss of detrusor smooth muscle density, and excessive connective tissue deposits were in line with reduced expression of calponin, smoothelin, and MYH11 [77]. Our study also highlighted the role of autophagy (a recycling process that degrades cellular components and provides energy) as a factor influencing the remodeling of SMCs, and the alteration of functionality in bladder smooth muscle tissue in NLUTD [77]. Furthermore, in a recent study, we demonstrated that SMCs from pediatric hostile bladders lose their characteristic disease phenotype in a three-dimensional (3D) microtissue culture and recover an improved contractile phenotype similar to that of healthy control subjects [78]. Overall, the actual mechanisms underlying the functional and structural changes in bladder SMCs in health and disease are not well understood. 

### 4.2. Urothelium

The umbrella cells of the urothelium express uroplakins and cytokeratin 20 (see Section 2.2) [79,80]. Furthermore, they can secret several factors, such as plasminogen activator and urokinase [37,38]. The urothelium can act as a sensor in response to mechanical stress/pressure via transient receptor potential channel (TRP) including vanilloid receptor TRPV1, vanilloid-like protein TRPV2, and TRPV4 [81]. The TRP proteins are also expressed in response to chemical stimuli. The TRPV1 is expressed by afferent nerves that form close contacts with urothelial cells, by urothelial cells themselves, and by SMCs [81,82,83]. In NLUTD samples from adults, changes in bladder urothelium protein expression were detected [50]. This study demonstrated a decrease in the expression levels of urothelial adhesion and junction proteins (E-cadherin, ZO-1, and UPK-3) and an increase in the expression of inflammatory proteins (tryptase, TNF-a, TGF-b) and cell proliferation (Ki-67) markers. These results indicate that NLUTD in SCI present with defects in regenerative function associated with chronic inflammation, increased apoptosis, and impairments in differentiation and barrier function [50]. Data on people with spinal dysraphism are not available.

### 4.3. ECM and Growth Factors

The composition of the ECM is dynamic and has a direct influence on the proliferation, differentiation, and homeostasis of SMCs [21]. Conditions linked to mechanical or chemical stress, such as NLUTD, can lead to inflammatory and fibroblastic responses in the ECM [35]. Structural proteins, such as collagen and elastin, as well as adhesive proteins, such as laminin and fibronectin, are produced and secreted by fibroblasts and myofibroblasts in the bladder wall [19]. These cells are also involved in wound healing, angiogenesis, and immune and inflammatory responses [19]. Among the paracrine factors that may influence fibroblast activity are platelet-derived growth factor (PDGF) and interleukin-6 (IL-6) [84]. As described above, fibrosis of the bladder wall is a crucial factor in NLUTD in children and adults. The remodeling of the ECM and the increased levels of transforming growth factor β1 (TGF-β1) are considered the underlying mechanisms of bladder fibrosis [85]. The TGF-βs are the cytokines overexpressed in fibrosis [86]. They bind to serine/threonine kinase receptors and act via phosphorylation in the SMAD2/3 pathway. Furthermore, IL-13, which is produced mostly by mast cells, stimulates the production of TGF-β1 and, thus, increases collagen production through the proliferation of fibroblasts and change to myofibroblasts [19,86]. The involvement of the TGF-β/SMAD signaling pathway in chronic bladder outlet obstruction leading to bladder fibrosis has been demonstrated [87]. In an animal model of juvenile rats with neurogenic bladder fibrosis, mimicking the main clinical feature of pediatric NLUTD, the early contribution of the TGF-β1 pathway has been shown [88]. This study demonstrated that the expression of TGF-β1 and its related protein SMAD2, along with collagen type III and collagen type I in bladder SMCs gradually increased over time after denervation of the bladder [88]. Inflammatory and fibroblastic responses further influence the fibroblast secretion of matrix metalloproteinases (MMPs) and tissue inhibitors of MMPs (TIMPs) [35]. The ongoing deposition of ECM increases the production of TIMPs, which are also associated with fibrotic changes in the bladder [35,89]. Activation of chemokines and pro-inflammatory cytokines, such as IL-1b and TNF-α, have been shown to be induced after SCI [90,91,92] in a manner similar to MMC. Specific studies in spinal dysraphism do not exist, but it is reasonable to assume that NLUTD dysfunction in the pediatric age group is subject to the same stresses on the bladder wall leading to the changes described.

The nerve growth factor (NGF) and its receptors expressed in the bladder are involved in urothelial cell proliferation [93,94]. Vaidyanathan et al. reported positive immunostaining for the p75 NGF receptor in the vesical urothelium of people with NLUTD [95]. Several studies in rodents and humans with NLUTD show that NGF affects the excitability of C-fibers in afferent nerves and bladder reflex activity [53]. Increased levels of NGF correlate with overactive bladder syndrome, indicating that NGF is an important factor in bladder physiology [39,96,97].

In summary, based on several studies performed on human MMC tissues and data acquired from animal models, the inflammatory cytokines TNF-α, IL-1b, IL-6, IL13, and TGF-β have been identified as potential contributors to the progressive damage in NLUTD tissue. Pro-fibrotic stimuli negatively impact bladder function and can lead to connective tissue remodeling, a reduction in the essential contractile proteins calponin, smoothelin, and MYH11, alterations in the regulation of growth factors, such as NGF, and an increase in MMPs, as well as a reduction in urothelial adhesion and junction proteins, such as E-cadherin, ZO-1, and UPK-3.

## 5. Outlook and Clinical Translation 

Congenital NLUTD causes life-long morbidity and strongly affects the quality of life of affected individuals. Progressive innervation dysfunction during pregnancy, developmental malformations of the bladder, pathophysiological changes due to detrusor overactivity, BOO, or inflammation lead to a thickening of the bladder wall with collagen deposits already in early childhood [7,8]. As a result, the bladder becomes less compliant and eventually fibrotic, losing its function of storing and emptying urine [87] (Figure 1). 

The current treatment approach for NLUTD in spinal dysraphism is primarily symptomatic. Intermittent catheterization ensures regular emptying, and the use of medication is aimed at improving the storage phase [1]. However, even with a therapy already started in infancy, a significant number of children experience progressive deterioration of bladder function, indicating that obstruction is not the only factor responsible for the morphological and functional changes, but that other influences need to be factored in. However, the contribution of the various underlying mechanisms and the exact pathogenesis remain unclear.

The most logical therapeutic approach would be the preservation of neurological function and, thus, prevention of NLUTD. Prenatal MMC repair aims to reduce neural damage and leads to an overall better outcome in these children [98,99]. Nevertheless, prenatal closure cannot prevent the development of NLUTD in the majority of children [100,101]. If the neurological function cannot be preserved in early development, efforts should be directed at preventing bladder deterioration, i.e., detrusor atrophy and fibrosis of the bladder wall. Although bladder fibrosis is a serious problem in various bladder diseases in adults as well, the underlying mechanism and risk factors have not yet been well identified. Possible targets to stop or reverse the process of bladder wall fibrosis are urgently needed. As mentioned above (see Section 4.3), different inflammatory cytokines have been identified as potential contributors to progressively increased ECM deposition and fibrosis. Targeting cytokine expression, such as TGF-β signaling, may represent a therapeutic option. Antifibrotic agents inhibiting the TGF-β pathway are known, but experience with bladder fibrosis is lacking [102].

Another mechanism that may contribute to detrusor failure is autophagy. This fundamental cellular degradation process is involved in the development of various pathologies. Recent studies have shown that autophagy is rapidly upregulated in myopathy, oxidative muscle stress, muscle inflammation, and muscle wasting [77]. We have shown that autophagy is also increased in bladder SMCs of children with advanced NLUTD and that it plays a role during disease development by allowing detrusor remodeling. These findings suggest that inhibition of autophagy may be another therapeutic strategy to prevent the deterioration of bladder function [77]. 

Various strategies should be pursued to prevent the progressive remodeling of the bladder wall. This complex process requires further studies to elucidate the underlying mechanism in order to identify effective treatments. Although acquired NLUTD and BOO share common pathophysiological mechanisms, it is important to acknowledge that the results of studies in adults or animal models have limited applicability to congenital NLUTD. Children present a particular challenge, as their diseases and capabilities evolve with age. An increased understanding of the signaling pathways involved in lower urinary tract development and function could also fill an important knowledge gap at the interface between basic science and clinical implications, leading to new opportunities for prenatal screening, diagnosis, and therapy [23].

## 6. Literature Search

For this non-systematic narrative review of the literature, the PubMed NCBI and Google Scholar electronic databases were searched for relevant peer-reviewed original or review articles. The search results were evaluated by the authors, reference lists were explored, and the literature was checked for cross-references. The literature search was restricted to English-language articles that were available in full text.

## Figures and Tables

**Figure 1 ijms-24-03692-f001:**
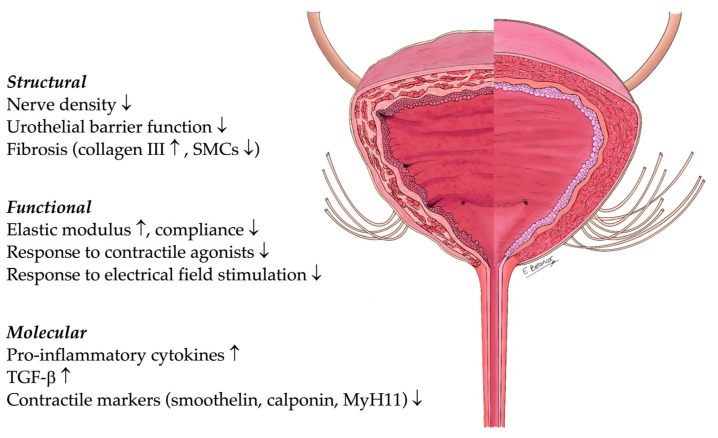
Structural, functional, and molecular changes in congenital NLUTD. The split image represents a healthy bladder on the right side and a NLUTD bladder on the left side. Abbreviations are as follows: NLUTD, neurogenic lower urinary tract dysfunction; SMCs, smooth muscle cells; TGF-β, transforming growth factor β; MyH11, myosin heavy chain 11.

## Data Availability

No new data were created or analyzed in this study. Data sharing is not applicable to this article.

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
