# Peer review of "Neurogenic Lower Urinary Tract Dysfunction in Spinal Dysraphism: Morphological and Molecular Evidence in Children"

_ijms, 2023, doi:10.3390/ijms24043692_

Round 1

Reviewer 1 Report

The authors present an interesting scoping review highlighting the evidence on morphology and molecular data related to neurogenic lower urinary tract dysfunction (NLUTD) in children with spinal dysraphism. While the overall work is very comprehensive, there are several queries which should be attended to before this reviewer can recommend the acceptance of this work.

In general:

Please refrain from using the term “neurogenic bladder” throughout the entire manuscript (incl. abstract) as this term is outdated. Please refer to official International Continence Society (ICS) terminology [https://www.ics.org/members/shop/icsstandards2023]. The author should either use neurogenic lower urinary tract dysfunction (NLUTD) or wherever appropriate.

Please refer to the intended age group, as currently, the authors switch between using different term: either utilize children, adolescents or adults wherever appropriate

Please consider favourable terms when addressing affected individuals, i.e. Lisa A. Harvey. Spinal Cord. 2019 Apr;57(4):257. [https://pubmed.ncbi.nlm.nih.gov/30952990/]

In detail:

Lines 26-30: There is a mix up with the first sentence, probably as copy and paste issue.

Line 31: please use “neurogenic lower urinary tract dysfunction” and NLUTD but not “(hereafter referred to as neurogenic bladder)” as aforementioned.

Lines 38-39: Since the authors use weeks throughout the manuscript when referring to time of development, the should also use it here, i.e. during the 3rd and 4th week of gestation.

Line 45: If referring to “pathophysiological changes due to detrusor overactivity“ in the contest of an underlying neurological disorder, then the authors should refer to it as “neurogenic detrusor overactivity (NDO)”, since detrusor overactivity (DO) alone (i.e. without an underlying neurological disorder) does not have to have a “pathophysiological” factor, such as in idiopathic conditions, e.g. overactive bladder (OAB) syndrome with DO.

Line 46: “bladder-sphincter dysfunction” is not specific, please utilize IC terminology, hence detrusor sphincter dyssynergia (DSD).

Line 48-49: “Upper and lower motor neuron lesions and autonomic dysfunction lead to a loss of 48 coordinated micturition” rather ‘Impairment of the central (CNS), peripheral (PNS), and autonomic (ANS) nervous system can result in NLUTD.’

Line 50: Please rephrase “bladder dysfunction”

Line 56 and throughout the manuscript: Please rephrase “hostile” as this term does not fit in the context of this research.

Line 57: With respect to “The consequences are either chronic overdistension of the detrusor with myogenic failure or detrusor hypertrophy due to increased outflow resistance during micturition.”, the authors should consider the development of bladder outlet obstruction (BOO) in adults, which first leads to a detrusor hypertrophy in order to compensate outflow resistance (e.g. prostate enlargement in older men or in those with DSD). If not treated appropriately, detrusor function will deteriorate, finally result in chronic overdistension of the bladder along with diminished voiding and increasing post-void residual long-term. Thus, the authors should emphasize a bit stronger on the differences between children with NLUTD, e.g. MCC, and adults.

Lines 58-59: The authors should provide (a) reference(s) for “Progressive changes lead to severe functional impairment with a loss of bladder wall elasticity and high intravesical pressure.”

Lines 64 / 134: Please consider to rephrase “bladder dysfunction”

Lines 86-87: “Bladder wall fibrosis leads to stiffening of the bladder wall, which increases intravesical pressure and exacerbates renal injury [2, 10].” Might be worthwhile to include the fact that bladder compliance will be impaired long-term, which could result in a low-compliance bladder.

Line 121: “has poor compliance” rather use low compliance in line with ICS terminology. “Compared to healthy bladders” rather refer to the condition in healthy control subjects (in particular specify if you refer to children, adolescents or adults).

Lines 132-133: “dyscoordination of the detrusor and the bladder sphincter”, there is no such thing as a bladder sphincter. There are two sphincters of urethra (internal and external). Please refer to the correct terminology with regards to the urethral sphincter (internal, external, or both).

Lines 135-148: This paragraph lacks the association of DSD and NDO/BOO with respect to the presented context. Currently, it appears that DSD has no influence on that matter (please see Yu and Kuo. Int Urol Nephrol. 2022 Aug;54(8):1815-1824. and Alatas et al. J Pediatr Neurosci. 2020 Jul-Sep; 15(3): 220–223).

Lines 173-178: With respect to reference 46 (i.e. A. Piaseczna-Piotrowska, M. Dzieniecka, E. Samolewicz, D. LeÅ›niak, and A. Kulig, "Distribution of interstitial cells of Cajal in 649 the neurogenic urinary bladder of children with myelomeningocele," (in eng), Adv Med Sci, vol. 58, no. 2, pp. 388-93, 2013, doi: 650 10.2478/ams-2013-0002.), this reviewer questions the control group described in the paper as “healthy controls”. Ten out of 16 in this group had a ‘Combined heart anomaly’ while the other six had other organ dysfunction leading to their death within 3 days. Thus, the reviewer feels that reference 46 is inadequate to distinguish the distribution of ICCs between children with MCC and those without NLUTD. Since the literature on that matter only highlights Ref 45 and 46 on PubMed, The authors might just comment on the fact that the control group labelled as healthy in Ref 46 was in fact not really healthy, which can bias the comparison between both, i.e. no statistical differences.

Lines 194-198: “Normal micturition is controlled by neuronal circuits in the spinal cord and brain via sympathetic and parasympathetic pathways. Cephalic centers coordinate the activity of spinal cord nuclei, that provide efferent innervation via ganglionic relay stations to the bladder. Afferent input by the bladder takes the same peripheral pathway back to the spinal cord and then through the ascending spinocephalic tract to sensory brain centers [17, 36].” Should be corrected and rephrased (please see Fowler, Griffiths, and William C. de Groat. Nat Rev Neurosci. 2008 Jun; 9(6): 453–466. and Panicker, Fowler, Kessler. Lancet Neurol. 2015 Jul;14(7):720-32.).

Lines 204-207: Might be worthwhile to highlight the difference between reflex voiding in children vs. SCI (please see Fig. 7 in Fowler, Griffiths, and William C. de Groat. Nat Rev Neurosci. 2008 Jun; 9(6): 453–466).

Line 207: “internal and external bladder sphincter” please rephrase as aforementioned and specify thereafter

Line 209: Rather NDO than “neurogenic overactive bladder”.

Lines 267-270: “As described above a hostile neurogenic bladder due to detrusor overactivity and/or BOO can lead to fibrosis of the bladder wall. The biomechanical consequences are decreased compliance and an increased elastic modulus of the bladder wall [18].” Please rephrase as commented on for lines 135-148.

Author Response

We would like to thank the Reviewer for the invested time, the more than valuable input and thus for the support to improve our manuscript. 

We addressed the Reviewers input as followed (see attached Word and track change in full text manuscript)

Author Response

We would like to thank the Reviewer for the invested time and for their support to improve this manuscript. 

We kindly addressed the Reviewers comments in the attached Word File.
